# Running behaviors, motivations, and injury risk during the COVID-19 pandemic: A survey of 1147 runners

**Alexandra F. DeJong**⬚*◉, **Pamela N. Fish**◉, **Jay Hertel**◉

University of Virginia Exercise and Sport Injury Laboratory, Charlottesville, Virginia, United States of America

◉ These authors contributed equally to this work.

* afd4au@virginia.edu

**Data Availability Statement:** Due to ethical restrictions, the data underlying this study may not be shared publicly. Data are available from the University of Virginia Ethics Committee (contact via Jeffrey Monroe, mjm6ny@virginia.edu) for

## Abstract

The COVID-19 pandemic has influenced activity behaviors worldwide. Given the accessibility of running as exercise, gaining information on running behaviors, motivations, and running-related injury (RRI) risk during the pandemic is warranted. The purpose of this study was to assess the influence of the COVID-19 pandemic on running volume, behaviors, motives, and RRI changes from the year prior to the pandemic to the timeframe during social isolation restrictions. Runners of all abilities were recruited via social media to complete a custom Qualtrics survey. Demographics, running volume, behaviors, motivations, and injury status were assessed for the year prior to the pandemic, and during social isolation measures. Descriptive statistics and Student's t-tests were used to assess changes in running outcomes during the pandemic. Logistic regressions were used to assess the influence of demographics on running behaviors and injury. Adjusted RRI risk ratios were calculated to determine the odds of sustaining an injury during the pandemic. Alpha was set to .05 for all analyses. A total of 1147 runners (66% females, median age: 35 years) across 15 countries (96% United States) completed the survey. Runners reported increased runs per week (Mean Difference with Standard Error [MD]: 0.30 [0.05], p < .001), sustained runs (MD: 0.44 [0.05], p < .001), mileage (MD: 0.87 [0.33], p = .01), and running times of day (MD: 0.11 [0.03], p < .001) during the pandemic, yet reported less workouts (i.e. sprint intervals; MD: -0.33 [0.06], p < .001), and less motives (MD [SE]: -0.41 [0.04], p < .001). Behavior changes were influenced by running experience and age. There was 1.40 (CI: 1.18,1.61) times the RRI risk during the pandemic compared to prior to the social isolation period. The COVID-19 pandemic influenced runners' behaviors with increased training volume, decreased intensity and motivation, and heightened injury risk. These results provide insights into how physical activity patterns were influenced by large-scale social isolation directives associated with the pandemic.

## Introduction

Running is one of the most popular forms of physical activity worldwide, and is an easily accessible form of exercise as there are minimal equipment and sport structure requirements

researchers who meet the criteria for access to confidential data.

**Funding:** The authors received no specific funding for this work.

**Competing interests:** The authors have declared that no competing interests exist.

[1]. According to the most recent report from the International Association of Athletics Federation, running races attracted over 107.9 million racers across 70,000 events in 2019, and running popularity has grown by approximately 60% over the past decade [2]. Running offers extensive health benefits, including decreased risk of chronic diseases [3] and improved mental health [4, 5], making this form of exercise an appealing health behavior for the general population. Additionally, many runners may opt to train in groups, clubs, or teams, thus introducing a social aspect to the activity [6].

The severe acute respiratory syndrome coronavirus 2, or the COVID-19, pandemic has imposed a unique and sweeping demand worldwide with government directives requiring the public to perform self-isolation behaviors and limit interpersonal exposures to mitigate the spread of this deadly virus. These wholesale changes have led to gym and exercise training facility closures, termination of formal and informal group activities, and restrictions on parks and trails that disrupted the norms of the distance running community. The COVID-19 pandemic has additionally led to many race cancellations or postponements that inevitably will result in training changes for competitive athletes [7]. However, there is currently no information available on how the pandemic has influenced running training behaviors, particularly in regards to running volume, intensity, training surfaces, and motives for engaging in running activities.

Another cogent concern associated with the COVID-19 pandemic is how the resultant shifts in running behavior and training schedules will influence the rates of running-related injury (RRI). Despite the aforementioned health benefits and motives associated with running as exercise, RRI's have long posed a substantial burden on the running community. Previous epidemiological research studies have found RRI incidence rates are as high as 90% of the running population, with the majority localized to the lower extremity [8, 9]. Up to 75% of injuries have additionally been categorized as overuse or recurrent pathologies primarily attributed to training errors [10, 11], such as sudden increases in running volume and intensity [12, 13]. The COVID-19 presents a unique external pressure on the running community and is likely to affect the injury occurrence in this population with necessary training adaptations. As such, interpreting how the pandemic is currently influencing running behaviors and RRI rates is critical for the greater community and for health care professionals treating runners in clinical settings. Data on RRI's would help clinicians prepare for patient volumes following re-opening and lifted restrictions on social isolation measures, and help to inform return to activity and injury prevention programming by assessing potential detraining or over-training indicators [12]. Clinicians may also be able to use information on running behaviors and running motivations to best inform future telemedicine programs [6].

The purpose of this study was to assess the effects of the COVID-19 pandemic on running behaviors among adult male and female runners of all experience and participation levels. Specifically, we aimed to assess running volume, running behaviors, motives for running, and RRI change from the year prior to the pandemic to the timeframe during social isolation restrictions. We hypothesized that overall, runners would present with increased running volume due to the accessibility of running as exercise, coupled with decreased running intensity due to changes in training goals and lack of access to tracks or training gyms. We additionally hypothesized that runners' motivations would be decreased during from before to during the pandemic. We anticipated that changes in running behaviors and motives would be dependent upon participant demographics, including age, sex, geographical location, and experience levels. In terms of injury risk, we hypothesized that there would be more RRI's localized to the lower extremity during the pandemic compared to before social isolation efforts due to acute changes in training.

## Materials and methods

### Participants

Adult male and female runners were recruited via social media outlets to complete an online survey (Qualtrics Labs Inc.). Participants were required to be at least 18 years of age, and either currently running or had been running within the last year at any participation and experience level. All respondents provided written consent to complete the survey prior to participation, and the study protocol was approved by the University of Virginia Institutional Review Board for Social and Behavioral Sciences IRB-SBS #3677.

### Survey instrument

The survey was developed in English by two researchers, and piloted among a group of 10 runners of varying age levels to determine face validity and refine the questions to improve clarity. The survey took approximately 10 minutes to complete, and the main components of the survey included participant demographics, running volume, behaviors, motives, and RRI's in the year prior to the COVID-19 pandemic, and running volume, behaviors, motives, and RRI's during the COVID-19 pandemic (S1 File).

The survey included demographic questions regarding age, biological sex, geographical location, and running experience. The remaining questions were posed both in the context of the year prior to social distancing restrictions in participants' geographical region due to the COVID-19 pandemic, and in the time during social distancing restrictions in participants' geographical region due to the COVID-19 pandemic. Only the year prior to the pandemic was used to minimize recall bias and reflect running training without capturing recent fluctuations in training cycles. Running volume was assessed by asking the participants' typical number of total runs per week, number of sustained runs per week, number of workouts per week (i.e. speed intervals, fartleks, tempo runs, hill repetitions, etc.), weekly mileage, and number of cross-training activities per week (i.e. strength training, cycling, swimming, yoga). Running behaviors were assessed by asking the participants' typical running pace during sustained runs and workouts, primary running locations (indoors, outdoors, both), use of technology to track runs, and typical time of day for training (early morning [5-7AM], mid-morning [8-10AM], midday [11AM-1PM], early afternoon [2-4PM], afternoon [5-7PM], evening [8-10PM], night [11PM-4AM]).

Running motives were assessed using a checklist for all reasons that the participants felt were applicable, and included exercise/fitness, competition/races, socialization, stress relief, enjoyment/pleasure, and to occupy free time. Several additional questions were included when assessing running behaviors and motivations during the COVID-19 pandemic to assess how much participants subjectively felt their training had changed on a 9-point scale ranging from "increased a great deal" to "decreased a great deal", and how concerned they were about their overall training and training goals on 5-point scales ranging from "very concerned" to "not concerned at all". Participants were also offered an optional open-ended section to provide any comments on how the COVID-19 pandemic affected their running that was not captured in the structured questions.

To assess RRI status, participants were asked if they had suffered from any RRI's (yes/no). We did not define RRI explicitly in the survey, however this was left intentionally broad to capture all injury data and became more specific in subsequent questions. Specifically, follow-up questions asked about the number of injuries they incurred, length of time taken off from running due to injury, length of time running training was modified due to injury, and chart to designate which body parts were affected (toe, foot, ankle, lower leg, knee, thigh, hamstring,

hip, groin, abdomen, low back) by injury type (sprain [ligament], strain [muscle], fracture [broken bone], other [explain]).

## Procedures

Adult runners were recruited to complete the online survey as a sample of convenience using a snowball sampling strategy. The survey link was initially distributed by the research team through personal and laboratory social media platforms (Twitter, Facebook, Instagram, LinkedIn, National Athletic Trainers' Association GATHER webpage). Others were encouraged to share the link via their own social media accounts to forward the survey to others who may qualify and be interested in participating. The link was also shared via email to other researchers and running club coordinators. Recruitment originated in the United States; however, runners were encouraged to participate regardless of geographical location. The survey was available from May 4th to June 4th of 2020 to capture the time period during peak social isolation restrictions in North America.

**Data processing.** Only complete responses were included in analyses. In order to assess changes in training volume variables, the reported values pertaining to behaviors in the year prior to the pandemic were subtracted from outcomes during the pandemic. In order to prepare for logistic regression analyses, demographic and running behavior data were binned as follows: age (18–25, 26–35, 36–45, 46–55, 56+ years), experience (0–3, 4–10, 11–15, 16–20 + years), and geographical location (US East Coast, US Mid-West, US West Coast, UK and Ireland, Canada, Other Regions). Running behavior outcomes were categorized as increased, decreased, or no change within 1 unit when assessing the change pre- to during the pandemic in number of total runs, sustained runs, workouts, cross-training, motives, and running times of day. Mileage was categorized in a similar manner within 10 miles per week, and pace within 30 seconds.

## Statistical analyses—Running behaviors and motives

Descriptive summary statistics were used to assess participant demographics, including biological sex, age, running experience in years, and geographical location. Descriptive statistics were additionally used to assess reported running behaviors prior to and during the pandemic, including number of total weekly runs, sustained runs, workouts, cross-training, weekly mileage, running pace, and use of technology to track runs. Histograms were used to visually assess data for normality, and the outcomes were observed to be normally distributed, supporting further analysis approaches. Student's t-tests were subsequently used to determine if there were statistically significant differences in numbers of reported runs (including run subtypes), number of motives, number of running time(s) of day, weekly mileage, and running pace during the pandemic as compared to prior behaviors. Alpha was set *a priori* to .05 for all analyses. Running motives, typical running time(s) of day, and running locations were evaluated by response type prior to and during the pandemic.

Multivariate logistic regression analyses were used to assess the influence of demographic factors on running behaviors during the pandemic compared to before. Demographic factors were first assessed in isolation and considered for inclusion into the final regression model were biological sex, and the binned age, years of running experience, the interaction between age and sex, the interaction between age and experience, and geographical location factors. Running behaviors included the binned change in number of total runs, sustained runs, workouts, motives, running times of day, and mileage. Cross-training and pace were not included into the model given that the Student's t-tests comparisons were not significantly different. Preliminary analyses reflected that there was not a significant age by sex interactions nor an

age by experience interaction for any outcome and thus were not included in the overall model. Additionally, preliminary results reflected that there were not enough observations in the UK/Ireland, Canada, and other geographic categories to appropriately run the analyses, and subsequently only US regions were included. The final logistic regression models assessed the influence of age, sex, running experience, and US geographical regions on running behaviors.

## Statistical analyses—Running-related injuries

Descriptive analyses were used to summarize the number of RRI's, location of injury, and length of time training was affected due to injury prior to and during the pandemic among the participant pool that reported RRI's. To assess RRI risk during the pandemic compared to before, an adjusted injury risk ratio was calculated using Eq 1. Given that social isolation took place primarily over three months, the percentage of injury prior to the pandemic was divided by four, and put into the model.

$$Injury\ Risk\ Ratio\ =\ \frac{([Adjusted\ Injuries\ Prior\ to\ Pandemic/Total\ Respondents]*100)}{([Injuries\ During\ Pandemic/Total\ Respondents]*100)} \quad (1)$$

Binary logistic regression analysis was used to assess the influence of demographic and training factors on injury occurrence during the pandemic. Demographic factors included into the regression model were biological sex, age, years of running experience, difference in mileage, difference in number of runs, and change in running location. Injury status in the past year was included as a covariate in the model given the influence of past injury on future injury risk.

## Statistical analyses—Response themes and additional outcomes

The short responses collected from the study procedures were assessed using an inductive qualitative approach to elucidate response themes. First, open coding was performed in R (R Development Core Team, 2011) to identify recurring words within all survey responses, in which participants' written responses were input into the coding platform and frequently appearing words were output along with the frequency count for each of these words. Two reviewers discussed open coding outcomes to help categorize responses into themes, and then independently evaluated all responses to label responses based upon the open coding results. The labeling process was continued until all responses were categorized into the most appropriate thematic bins, and the reviewers then re-convened to compare labels and resolve any discrepancies. In the event of any labeling discrepancies, the results were discussed amongst the study team until a consensus was reached. Finally, descriptive statistics were used to assess participants' perceptions of the pandemic's influence on their running training and goals.

## Results

### Running behaviors and motives

There were a total of 1147 complete responses recorded across 46 states and 15 countries, the majority of which coming from the United States due to the convenience sampling methods (Fig 1A and 1B). Respondents were primarily females (66%), the median age of respondents was 35 years, and the median length of running experience was 7–8 years (Fig 2). Descriptive outcomes and outcomes of the Student's t-tests comparing running volume, pace, running behaviors, and use of running tracking technology prior to and during the pandemic can be found in Table 1. Overall, total number of runs, number of sustained runs, mileage, and

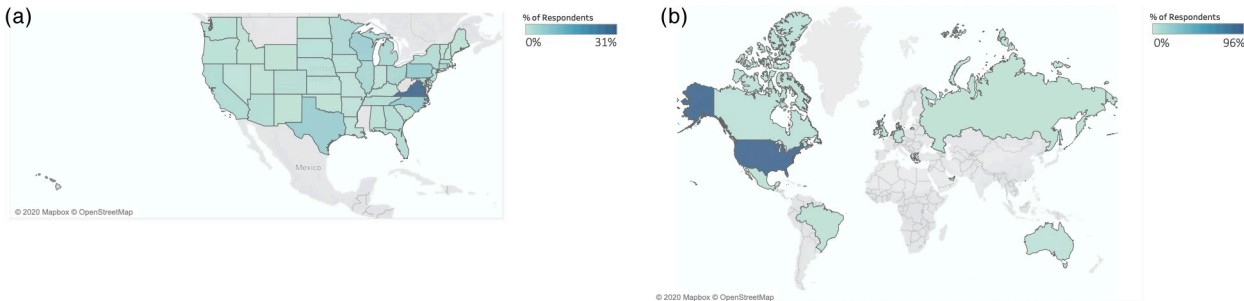

**Fig 1.** Responses (A) by state and (B) by country. Full List of Respondents—Australia (<1%), Brazil (<1%), Cayman Islands (<1%), Canada (1%), Denmark (<1%), Germany(<1%), Greece(<1%), Hong Kong(<1%), Ireland (<1%), Mexico (<1%), Netherlands (<1%), Russia (<1%), United Arab Emirates (<1%), United Kingdom (1%), United States (96%). List of missing states: Alaska, Mississippi, Montana, West Virginia.

running times of day significantly increased during compared to before the pandemic, however the total number of workouts per week and running motives significantly decreased (Table 1, Fig 3A and 3B). Changes in activity by state and by country can be found in S1A–S1F Fig.

When examining shifts in running motives during the pandemic, there was a decrease in responses for competition/races and socialization as driving factors for running participation, while there were more responses that participants were motivated to run to occupy free time (Fig 3C). Less runners reported exercising early in the morning (5-7AM) and the early evening

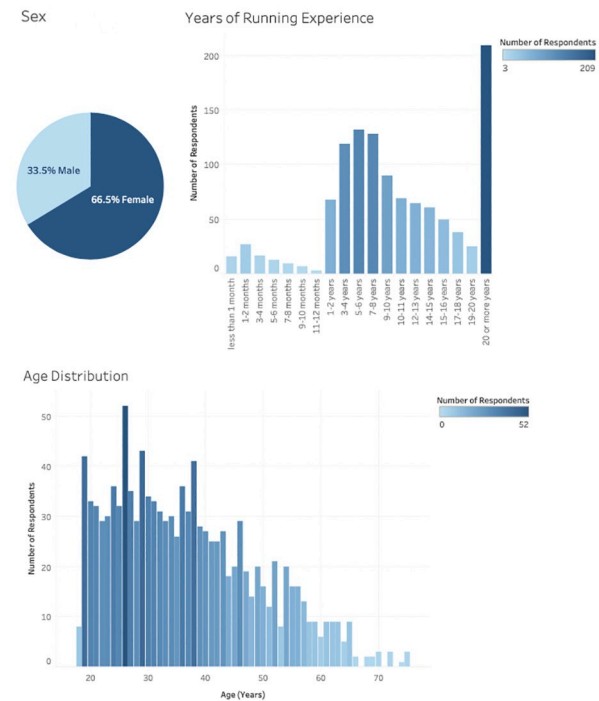

**Fig 2. Descriptive outcomes for respondents by sex, years of running experience, and age.** Percentage of male and female respondents, extent of running experience, and age distribution of respondents depicted across the figure graphics. Greater number of respondents per category is depicted in darker shades of blue for experience and age histograms.

**Table 1. Mean outcomes and student's t-test comparisons on running behaviors before and during the pandemic.**

| | Before the Pandemic | During the Pandemic | Mean Difference (Standard Error) | P-Value |
|---|---|---|---|---|
| | Mean (95% CI) | Mean (95% CI) | | |
| **Total Runs Per Week (N)** | 4.27 ± 1.93 | 4.56 ± 1.92 | 0.30 (0.05) | < .001* |
| **Sustained Runs Per Week (N)** | 3.25 ± 1.78 | 3.69 ± 1.90 | 0.44 (0.05) | < .001* |
| **Workout Runs Per Week (N)** | 1.60 ± 1.5 | 1.93 ± 1.54 | -0.33 (0.06) | < .001* |
| **Cross-Training Per Week (N)** | 3.17 ± 2.09 | 3.09 ± 1.97 | 0.08 (0.07) | 0.25 |
| **Weekly Mileage (mi)** | 24.63 ± 18.36 | 25.49 ± 18.38 | 0.87 (0.33) | .01* |
| **Pace (min/mi)** | 9:48 ± 1:51 | 9:24 ± 1:54 | -0:24 (0:29) | 0.35 |
| **Running Motives (N)** | 3.05 ± 1.73 | 2.65 ± 1.53 | -0.41 (0.04) | < .001* |
| **Typical Running Time Blocks (N)** | 1.71 ± 0.92 | 1.82 ± 1.04 | 0.11 (0.03) | < .001* |

Abbreviations: CI, confidence interval; N, number; mi, mile; min, minute.

*Significant at $p < .05$.

(5-7pm), but increased activity mid-day (11AM-1PM) during the pandemic compared to prior running behaviors (Fig 3D). Finally, respondents reported running substantially more outdoors than indoors during the pandemic than before (Fig 3E).

The results of the regression analyses assessing the influence of demographic factors on running behaviors can be found in Table 2. The only significant finding for sex in the model was that males were less likely to decrease (Odds Ratio with 95% Confidence Interval [OR]: 0.66 [0.47,0.93) or increase (OR: 0.47 [0.34, 0.64]) their weekly mileage compared to females. Otherwise, the key factors influencing running behavior changes were related to running experience and age. Notably, runners with 0–3 years of running experience were significantly less likely to decrease their number of runs per week than maintain their running volume compared to runners with 4–10 and 11–15 years of experience ($OR_{0-3 \text{ vs. } 4-10}$: 0.60 [0.42, 0.88]; $OR_{0-3 \text{ vs. } 11-15}$: 0.62 [0.38,0.99]), and were less likely to increase their number of sustained runs per week than maintain running habits when compared to runners with 4–10 and 11–15 years of experience ($OR_{0-3 \text{ vs. } 4-10}$: 0.66 [0.45, 0.97]; $OR_{0-3 \text{ vs. } 11-15}$: 0.49 [0.29,0.82]). However, runners with 0–3 years of experience also were more likely to decrease their overall weekly mileage compared to runners with 4–10 years of experience (OR: 1.80 [1.17, 2.76]). Runners with 0–3 years of experience were also significantly less likely to decrease their number of reported workouts per week when compared to runners with 11–15 and 16–20+ years of experience ($OR_{0-3 \text{ vs. } 11-15}$: 0.43 [0.22, 0.83]; $OR_{0-3 \text{ vs. } 11-15}$: 0.39 [0.20, 0.75]), and less likely to increase their number of workouts compared to 16–20+ years of experience during the pandemic (OR: 0.25 [0.08, 0.73]). Finally, runners with 0–3 years of experience were less likely to increase their running motives compared to all other runner groups ($OR_{0-3 \text{ vs. } 4-10}$: 0.35 [0.20, 0.60]; $OR_{0-3 \text{ vs. } 11-15}$: 0.20 [0.08, 0.48]; $OR_{0-3 \text{ vs. } 11-15}$: 0.23 [0.10, 0.53]).

When assessing the effects of age in the model, younger runners ages 18–25 were more likely to increase number of runs per week compared to runners ages 46–55 (OR: 1.44 [1.23, 1.86]). Additionally, runners ages 18–25 were significantly less likely to decrease their overall weekly mileage compared to older runner groups during the pandemic ($OR_{18-25 \text{ vs. } 26-35}$: 0.60 [0.38, 0.93]; $OR_{18-25 \text{ vs. } 36-45}$: 0.57 [0.35, 0.92]; $OR_{18-25 \text{ vs. } 46-55}$: 0.40 [0.22, 0.71]; $OR_{18-25 \text{ vs. } 56+}$: 0.50 [0.26, 0.97]). Finally, runners ages 18–25 were significantly less likely to decrease their number of workouts ($OR_{18-25 \text{ vs. } 26-35}$: 0.53 [0.32, 0.86]), and training times of day during the week ($OR_{18-25 \text{ vs. } 26-35}$: 0.30 [0.12, 0.73]; $OR_{18-25 \text{ vs. } 36-45}$: 0.35 [0.14, 0.89]) during the pandemic.

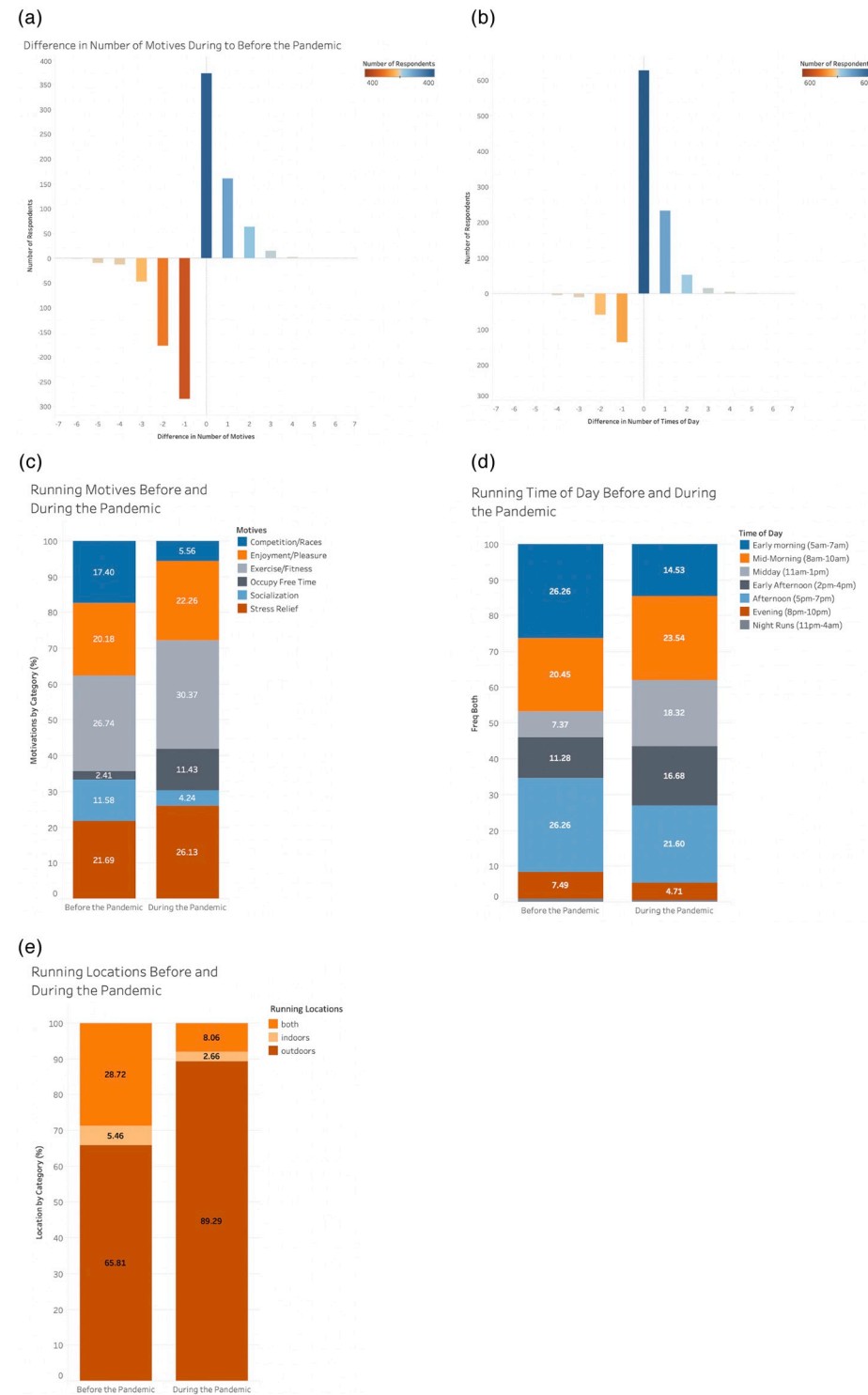

**Fig 3.** (A) Difference in number of running motives and (B) running times of day during the pandemic. (C) Motives for running and (D) running times per day before and during the pandemic by percentage responses. (E) Running locations before and during the pandemic by percentage responses. Differences in numbers of (A) running motives and (B) running times of day represented in the histograms, with increases in outcomes depicted in blue, and decreases depicted in red, and increased number of occurrences represented with darker color shades. Percentage responses for (C) running motives, (D) running times per day, and (E) running locations before and during the pandemic are depicted within the stacked bar plots.

**Table 2. Logistic regression outcomes assessing the influence of demographic factors on running behavior changes during the pandemic.**

| Predictor | Comparison | Total N Runs Odds Ratio (95% CI) | | Sustained Runs Odds Ratio (95% CI) | | Workouts Odds Ratio (95% CI) | | Motives Odds Ratio (95% CI) | | Times of Day Odds Ratio (95% CI) | | Mileage Odds Ratio (95% CI) | |
|---|---|---|---|---|---|---|---|---|---|---|---|---|---|
| | | ↓ vs. No Change | ↑ vs. No Change | ↓ vs. No Change | ↑ vs. No Change | ↓ vs. No Change | ↑ vs. No Change | ↓ vs. No Change | ↑ vs. No Change | ↓ vs. No Change | ↑ vs. No Change | ↓ vs. No Change | ↑ vs. No Change |
| Age | 18–25 vs. 26–35 | 1.2 (0.78, 1.84) | 0.81 (0.52, 1.27) | 0.98 (0.55, 1.72) | 0.98 (0.64, 1.50) | **0.53 (0.32, 0.86)** * | 1.02 (0.35, 2.96) | 1.12 (0.72, 1.73) | 1.06 (0.59, 1.89) | **0.30 (0.12, 0.73)** * | 0.74 (0.35, 1.54) | **0.60 (0.38, 0.93)** * | 0.79 (0.51, 1.21) |
| | 18–25 vs. 36–45 | 0.88 (0.56, 1.41) | 0.61 (0.37, 1.01) | 0.73 (0.39, 1.36) | 0.82 (0.52, 1.29) | 0.60 (0.36, 1.01) | 1.93 (0.70, 5.33) | 1.02 (0.64, 1.62) | 0.60 (0.29, 1.21) | **0.35 (0.14, 0.89)** * | 0.92 (0.44, 1.95) | **0.57 (0.35, 0.92)** * | 0.73 (0.46, 1.17) |
| | 18–25 vs. 46–55 | 1.02 (0.60, 1.73) | **1.44 (1.23, 1.86)** * | 0.58 (0.27, 1.25) | 0.68 (0.40, 1.18) | 0.56 (0.28, 1.12) | 1.91 (0.55, 6.67) | 1.02 (0.60, 1.72) | 0.47 (0.18, 1.22) | 0.38 (0.13, 1.10) | 0.48 (0.18, 1.23) | **0.40 (0.22, 0.71)** * | 0.70 (0.42, 1.18) |
| | 18–25 vs. 56+ | 0.77 (0.39, 1.54) | 1.15 (0.60, 2.22) | 0.96 (0.42, 2.18) | 0.49 (0.24, 1.01) | 0.53 (0.22, 1.27) | 3.47 (0.92, 13.13) | 1.11 (0.60, 2.07) | 0.48 (0.13, 1.78) | 0.33 (0.08, 1.35) | 0.41 (0.13, 1.30) | **0.50 (0.26, 0.97)** * | 0.43 (0.22, 0.84) |
| Sex | Male vs. Female | 1.04 (0.75, 1.43) | 1.19 (0.82, 1.72) | 0.75 (0.49, 1.14) | 1.06 (0.76, 1.46) | 1.30 (0.87, 1.96) | 1.84 (0.83, 4.09) | 1.35 (0.97, 1.87) | 1.22 (0.72, 2.06) | 1.70 (0.81, 3.58) | 1.31 (0.77, 2.23) | **0.66 (0.47, 0.93)** * | **0.47 (0.34, 0.64)** * |
| Years Running Experience | 0–3 vs. 10-Apr | **0.6 (0.42, 0.88)** * | 0.94 (0.62, 1.41) | 0.85 (0.51, 1.43) | **0.66 (0.45, 0.97)** * | 0.90 (0.57, 1.41) | 0.42 (0.18, 1.00) | 1.28 (0.86, 1.92) | **0.35 (0.20, 0.60)** * | 1.35 (0.59, 3.11) | 1.03 (0.50, 2.14) | **1.80 (1.17, 2.76)** * | 0.90 (0.61, 1.32) |
| | 0–3 vs. 15-Nov | **0.62 (0.38, 0.99)** * | 0.64 (0.36, 1.14) | 0.59 (0.29, 1.17) | **0.49 (0.29, 0.82)** * | **0.43 (0.22, 0.83)** * | 0.42 (0.15, 1.21) | 1.04 (0.63, 1.72) | **0.20 (0.08, 0.48)** * | 1.39 (0.47, 4.17) | **2.55 (1.18, 5.51)** * | 1.43 (0.82, 2.49) | 1.27 (0.79, 2.02) |
| | 0–3 vs. 16–20+ | 0.79 (0.50, 1.23) | 0.68 (0.39, 1.19) | 0.73 (0.38, 1.40) | 0.84 (0.53, 1.33) | **0.39 (0.20, 0.75)** * | **0.25 (0.08, 0.73)** * | 1.14 (0.71, 1.83) | **0.23 (0.10, 0.53)** * | 1.72 (0.57, 5.17) | **2.70 (1.23, 5.89)** * | 1.56 (0.91, 2.65) | 1.10 (0.70, 1.74) |
| Location | US EC vs. US Mw | 0.79 (0.56, 1.11) | 1.00 (0.69, 1.44) | 1.51 (0.74, 1.78) | 1.05 (0.75, 1.46) | 0.94 (0.64, 1.39) | 0.61 (0.28, 1.35) | 1.28 (0.93, 1.76) | 1.17 (0.70, 1.95) | 0.89 (0.44, 1.82) | 0.73 (0.42, 1.29) | 1.03 (0.73, 1.47) | 1.12 (0.81, 1.55) |
| | US EC vs. US WC | 1.73 (0.87, 3.46) | 1.34 (0.56, 3.19) | 0.50 (0.11, 2.17) | 1.70 (0.84, 3.43) | 0.48 (0.14, 1.64) | 1.35 (0.30, 6.19) | 0.94 (0.42, 2.10) | 1.49 (0.49, 4.59) | 1.82 (0.51, 6.45) | 0.61 (0.14, 2.62) | 1.16 (0.52, 2.59) | 1.21 (0.58, 2.50) |

Table presenting the results of the logistic regression model. The odds ratios presented are the results when comparing the first listed group to the second, such that if the odds ratio is less than one that the first group listed was less likely to change behaviors, and conversely if the odds ratio is greater than one the first group listed was more likely to change. The odds ratio reference was no change in running behavior, and both likelihoods to increase or decrease running behaviors were assessed and presented in the table columns. Abbreviations: N, number; CI, confidence interval; US, United States; EC, East Coast; WC, West Coast.

*Significant at p < .05.

## Running-related injuries

410 participants (35.7%) reported a total of 634 injuries in the year prior to the pandemic (Average days off of running: 42±48; Average days of modified running: 60±63), while 144 (12.6%) reported a total of 219 injuries during the 3-month social isolation period of the pandemic (Average days off of running: 10±12; Average days of modified running: 18±34). Of the reported RRI's incurred prior to the pandemic, 63 participants reported injury during the pandemic (15.4%). Injuries by type and body part can be found in Table 3. While raw outcomes reflected a higher number of injuries prior to the pandemic, the 3-month adjusted injury risk

**Table 3. Injuries reported before and during the pandemic by injury type and body part.**

| | Sprain / Ligamentous | | Strain / Musculotendinous | | Fracture / Stress Fracture / Bony | | Other | | Total Injuries by Location | |
|---|---|---|---|---|---|---|---|---|---|---|
| | (N reported) | | (N reported) | | (N reported) | | (N reported) | | | |
| | Before | During | Before | During | Before | During | Before | During | Before | During |
| Toe | 6 | 0 | 2 | 2 | 7 | 1 | 2 | 1 | 17 (2.68%) | 4 / -2.44% |
| Foot | 33 | 11 | 57 | 10 | 14 | 4 | 8 | 6 | 112 (17.67%) | 31 (18.90%) |
| Ankle | 44 | 9 | 38 | 11 | 7 | 0 | 3 | 1 | 92 (14.51%) | 21 (12.80%) |
| Lower Leg | 4 | 2 | 59 | 14 | 20 | 2 | 22 | 8 | 105 (16.56%) | 26 (15.85%) |
| Knee | 25 | 4 | 51 | 22 | 0 | 0 | 22 | 15 | 98 (15.46%) | 41 / -25% |
| Thigh | 0 | 0 | 24 | 5 | 1 | 0 | 1 | 0 | 26 (4.10%) | 5 / -3.05% |
| Hamstring | 4 | 1 | 41 | 5 | 0 | 0 | 3 | 2 | 48 (7.57%) | 8 / -4.88% |
| Hip | 2 | 0 | 47 | 7 | 6 | 1 | 16 | 10 | 71 (11.20%) | 18 (10.98%) |
| Groin | 1 | 0 | 7 | 4 | 1 | 0 | 1 | 0 | 10 (1.58%) | 4 / -2.43% |
| Abdominals | 1 | 0 | 1 | 0 | 2 | 0 | 1 | 0 | 5 / -0.79% | 0 / 0% |
| Lower Back | 2 | 0 | 33 | 3 | 2 | 0 | 13 | 3 | 50 (7.89%) | 6 / -3.66% |

Abbreviations: N, number; Before, the year prior to the pandemic; During, the period of social isolation during the pandemic.

ratio demonstrated that there was 1.40 times the injury risk (Confidence Interval: 1.18,1.61) during the pandemic as compared to prior to the social isolation period. The logistic regression model only explained 1.66% of the variance, and none of the demographic factors included in the model were significant predictors of RRI during the pandemic when covarying for previous injury.

## Response themes and additional outcomes

Of the original 1147 respondent sample, 638 participants provided short responses. Based on the results of the open coding assessment, several notable key words were identified that aided in thematic response labeling (S1 Table). The major emergent themes were competition changes (610 instances), motivation (192 instances), well-being (69 instances), situational factors (181 instances), social support (121 instances), and resiliency (181 instances). Under the competition changes theme, two sub-categories emerged: race cancellations without alternative participation methods, and race postponements/virtual race alternatives. Under motivation, the two underlying categories identified were decreased motivation to continue running, and no effect on training goals due to established training habits. Under the well-being theme, the two emergent categories were increases in training to improve health, and alterations in training due to fear of injury occurrence. All other themes had a singular underlying category in responses, and response examples by theme categories can be found in Table 4.

The majority of participants (57.82%) perceived that their running training had increased ranging from a little to a great deal, while the remaining participants felt that their training had

**Table 4. Short response examples by response themes.**

| Response Theme | Category | Sample Responses |
|---|---|---|
| Competition Changes **(610 Instances)** | Race cancellations or uncertainty without alternative participation methods **(381 Instances)** | "I was planning on running a race in March 2020 which got cancelled. I was also planning on potentially running a race in fall 2020, but I haven't signed up nor do I plan to at the moment because of COVID-19." |
| | | "I had been training for a half which was ultimately postponed. No date has been announced yet so I can't set up a new schedule. . ." |
| | | "I had been training for 5 months and the race was canceled. I miss the spring 5K races. I'm concerned about the fall 5K races." |
| | | "I am concerned about the upcoming cross country and track seasons. Even if season happens, we will be severely restricted due to huge budget cuts. I am concerned I will have to pay to be a student athlete." |
| | | "I am supposed to start training in June for a marathon to be held in October. I am hesitant to start this high intensity training with the possibility that the race may be cancelled." |
| | Race postponements/Virtual race alternatives **(229 Instances)** | "Can't race in planned events but registered for 2 virtual [races]" |
| | | "Have done more virtual races." |
| | | "I'm participating in Virtual Events just-in-case my fall running events get canceled." |
| | | "The pandemic has canceled/post-poned a lot of my races, so I've opted to do some virtually (which is fine, but NOT the same). . ." |
| Motivation **(192 Instances)** | Decreased motivation to continue running due to lack of extrinsic motivators **(175 Instances)** | "Hard to set goals with no races" |
| | | ". . .I have fallen back to running 3–4 miles at most because I've lost the motivation to run without an upcoming race (since all have been canceled) and without friends to run with." |
| | | "The interruption to my training has been mental. I've struggled to find motivation to maintain my training, and I'm generally mentally healthy. This has just been different, and it has affected me more than I ever would have expected." |
| | | "I've mainly lost my motivation without having the social or competition aspect of running." |
| | No effect on training goals due to established training methods **(17 Instances)** | "My training has worked for me for years and I won't change it because of this one virus." |
| | | "I still have goals. I still have things I'm working towards. A break from racing doesn't change that." |
| | | "honestly hasn't changed running for me at all. . .[COVID] won't stop my love for running and training in general" |
| Well-Being **(69 Instances)** | Running as a means to improve health and wellness **(56 Instances)** | "I've become much more committed to regular runs in order to improve cardiovascular health, in order to improve my chances of recovery if I contact COVID-19" |
| | | "Just running for fun and fitness and increasing stretching and weight training." |
| | | "As someone who has health issues the year prior I saw this as an opportunity to regain lost fitness and still set goals without races. . ." |
| | Altered training habits due to fear and/or occurrence of injury and illness **(13 Instances)** | "The changes to my training are mostly injury related (still recovering from plantar fasciitis). . ." |
| | | "I am a little concerned about increased risk of injury due to more volume on roads." |
| | | "The biggest change I made was shortening my long run each week. I did this in order to avoid any overuse injury during the pandemic." |

(*Continued*)

**Table 4.** (Continued)

| Response Theme | Category | Sample Responses |
|---|---|---|
| Situational Factors (**181 Instances**) | Home or locally-imposed restrictions changing running training habits | "Being home 24/7 and having added responsibilities has made taking care of myself more difficult" |
| | | "At the start in March I was running 4–6 times a week. Then our county required face masks and I run on a walking path pushing a stroller. Due to that I stopped running." |
| | | "Due to schedule changes necessitated by the quarantine, I have been forces to conduct interval training on an assault bike in lieu of running." |
| | | "I am a resistance trainer at heart, but out of lack of options have had to increase my running dosage and frequency as all the gyms are closed." |
| | | ". . .The XXXX Trail is too busy to run responsibly. I now run almost exclusively on the road/sidewalk, which was never my preference. . ." |
| | | "I have more time to train because I no longer need to commute to work (40 min. each way)—I am currently working from home." |
| | | "I'm a terrible runner and always wanted to start, so COVID-19 has given me a chance to gradually build up my ability due to more free time and good weather." |
| | | "I changed from my normal scenic trail to a less populated area (sidewalks in a neighborhood instead of a running path along a river)" |
| | | "With two teenage girls distance learning and my normal work routine disrupted by educational closures, mileage went down dramatically." |
| Social Support (**121 Instances**) | Lack of social groups creating training changes | "I miss racing and having a goal race. I also miss big group runs." |
| | | "I miss having a tangible goal, and I REALLY miss my running groups and friends. Tougher to stay motivated when you can't run with other people." |
| | | "It is much harder to do speed work alone. I miss chasing my friends at the track and the motivation we provide each other." |
| | | "The lack of running partners makes long runs very difficult" |
| Resiliency (**181 Instances**) | Began/Maintained running with positive outlook | "Just working on getting better and enjoying the process" |
| | | "I am running more now than before!" |
| | | "I did not run regularly before COVID-19. I now run several times a week and set goals as well, goals that I have been meeting." |
| | | "I have been much more consistent with my running during this pandemic." |
| | | "I've taken time to stop marathon/ half marathon training focus and instead work on flat speed (mile to 5k training) and increased cross training knowing that both of these will benefit me in the long run to stay healthy and get faster in a time of races. This will translate to faster halves and fulls in the future. Plus it's fun to do something a little different! I think this change also helps mentally to fight any burnout." |
| | | "I have been able to get in quality training and actually have been able to run more mileage and harder workouts." |

Sample responses provided from the survey under response theme categories.

not changed (12.32%), or decreased to varying degrees (29.86%). About 20% of respondents reported feeling somewhat or very concerned about the effect the pandemic will have on their running training, and about 40% of respondents were somewhat or very concerned about their running goals.

## Discussion

The COVID-19 pandemic imposed a unique stress on daily functioning worldwide, and the results of this survey indicate that there has been a reactionary response in the running community. The primary hypotheses were mainly supported in that the runners sampled in this survey reported increased running volumes with decreased intensity, coinciding with

increased injury risk and alterations in running training motives. These findings may be of use to coaches when developing training programs and sports medicine clinicians in preparing for patient loads as social isolation restrictions become lifted.

## Running behaviors and injury risk

Overall, runners increased their number of runs per week, weekly mileage, and number of times of day they opted to run. We postulated that this response would be seen given the accessibility of running training and the physical and mental health benefits associated with this form of cardiovascular activity [3]. This response was perpetuated in the short answer health response themes, as individuals cited beginning or maintaining running to combat weight gain, maintain fitness, and protect against COVID-19 respiratory health complications [14]. Running motives additionally shifted away from social and competitive aspects of the activity, and towards stress relief, occupying free time, and fitness. We believe these underlying factors, along with access to facilities such as gyms and tracks reported in short answer responses, further coincide with the noted decrements in training intensity and fewer overall number motives for continuing training.

Tendency towards increased running volume but with decreased running intensity is critical information for coaches and sports medicine clinicians to consider during the transition back to normal training schedules. Sudden increases in running training intensity have been associated with acute lower extremity injuries, such as Achilles tendinopathies and gastrocnemius strains [13]. Our findings suggest that runners decreased their running intensities which may have reduced some acute injury outcomes in the pandemic timeframe. However, there is a potential that sudden re-introduction of high intensity training would result in acute injury risk. Therefore, an emphasis on gradual re-introduction to workouts and higher intensity training such as intervals and speed workouts should be considered during return to competitive running training given the noted decline in training intensity during social isolation. Knowledge of these running behavior changes will help coaches and sports scientists to create graded return to activity protocols bearing in mind progressive, cyclical periodization tactics to mitigate acute injury in the running community [15]. Although many runners reported an eagerness to return to racing and high intensity training in their short answer responses, runners should be informed of the risks acute training changes may imposed on their physical health.

In a similar vein, overuse injuries are frequently cited to occur with sudden training changes [12, 13, 16], akin to the documented increase in total number of runs, mileage, and sustained runs during the pandemic. Prevalent lower extremity overuse injuries are linked with training errors, such as running primarily on asphalt and high weekly running exposure [12, 13, 17, 18]. Not only was the injury risk higher among participants during the pandemic, but the majority of reported injuries were categorized as two of the most prevalent overuse RRI's: patellofemoral pain and medial tibial stress syndrome housed within the "other" injury response categories (Table 3). Further, locally-imposed restrictions resulted in a 23% increase in exclusive outdoor running training among participants (Fig 3E). Although these factors were not significant predictors for sustaining RRI, these factors should still be clinically considered, particularly in the upcoming year as running exposures continue during the gradual return to routine functioning.

Continued health monitoring among the running community is a necessary next step to determine how RRI epidemiology will shift in the upcoming years. As a note of caution, we did not explicitly ask participants about overuse and chronic injury categorizations, and instead kept the injury definition intentionally broad with greater details elucidated in the

injury types by body part chart. Further, as injury outcomes were assessed for the year leading up to the pandemic, there is a potential for recall bias in the reported outcomes. These concepts should be considered when interpreting the injury data; however, we believe the reported information provides valuable insights into injury outcomes during this unique time. Clinicians treating RRI's should be aware of the noted increase in time-adjusted overuse injury outcomes determined from this study sample to adequately prepare for patient volumes when clinics begin to re-open.

It was a surprising finding that cross-training activities did not significantly differ during the pandemic; we anticipated cross-training to decrease due to limited access to gyms and other training facilities. However, short-responses reflected that individuals opted to perform lighter intensity cross-training activities due to lack of access to heavier weights and machinery that were not adequately captured in this survey response. Key stakeholders should assess athletes' abilities to perform cross-training activities during the pandemic period, and the intensity of the exercises before returning to strenuous exercise.

## Personal and geographic factors

The largest demographic factor influencing running training was found to be age; younger runners were significantly less likely to decrease their mileage compared to all other age groups. We believe this is attributed to personal demands, particularly as runners in older age categories reported increased work and home-schooling demands. It was somewhat of a surprising finding that there was not a significant interaction between age and sex on running behaviors, nor between age and experience on running behaviors. We anticipated that female runners ages 36–45 would report lesser running volume due to increased familial demands, and that younger runners with more experience would be more resilient to social isolation changes in running behaviors due to less home-based demands and more routine training. However, experienced runners were less likely to decrease their weekly training volume regardless of age which we believe is reflective of established training plans, and females were more likely to change weekly mileage than males.

Unfortunately given the convenience sampling methods, there were not enough international responses to determine if isolation measures disproportionally influenced runners' behavioral responses by geographical location. An added layer of difficulty to interpretation of responses is that many social isolation decision measures were made at local regional levels and not on full state or country levels. Based on our sampling approach, we were unable to determine if there was a differential response based upon exact region, especially when considering differences such as rural versus urban habitancy which may have influenced the results. We found that there was not a significant influence of geographical region on running behaviors, although individuals did cite local closures and requirements of wearing masks during activity that hindered their running training.

## Implications for telemedicine

The results of this survey highlight several important opportunities for coaches and sports medicine clinicians to leverage technology that runners are already using to improve runners' motivations, aid in goal-setting, and mitigate injury risk during periods of remote interactions. As ~90% of runners reported utilizing technology to keep track of their runs, there is already a wealth of information available on habitual training behaviors such that remote interventions can be tailored and facilitated to meet runners' needs during this time.

Runners consistently reported decreases in social support as barriers to their running training during the pandemic, and short-responses emphasized missing community and team

encouragement to maintain running habits. Furthermore, the majority of runners reported concerns about their overall running goals during the social isolation period. While these were the main themes throughout runners' responses, there was a subgroup of runners that demonstrated resilience and alternative means of encouragement by completing virtual races and remote running clubs. Clinicians and coaches can utilize this information to help increase adherence to running and reduce the barriers to training in future periods of remote contact. Although the pandemic is distinct in that direct contact is not possible, there are still situations in which coaches and clinicians may not be able to directly come in contact with runners (i.e. off-season training, geographical location differences, etc.). Key stakeholders may consider performing motivational interviewing with runners to help elucidate individuals' needs for extrinsic motivating factors contributing to their own training. This would help introduce the opportunity connect runners with one another using virtual tracking to foster social support, and connect with virtual race opportunities to prevent losing competitive motivation [19]. In terms of maintaining runners' goals or creating tangible running goals, technology-based coaching has been found to increase adherence to workout schedules and more regular training that would help prevent sudden training changes as seen in this study [20].

The injury outcomes from this survey highlight the importance of incorporating telehealth initiatives to inform runners of the risks of training volume and overuse injuries as preventative measures. While direct face-to-face patient-clinician interactions are being gradually re-introduced for non-emergent medical conditions, clinicians should become more adept to leveraging technology to reach their patients. The findings from this survey support sharing general training information to runners to mitigate spikes in injury epidemiology over the upcoming months to years, with a particular emphasis on training dosage. These plans would be particularly beneficial for runners with some limited running experience (4–10 years), as this runner group was found to be more susceptible to training changes. These plans may also be helpful for novice runners who reported a decline in weekly running mileage during the pandemic; this group may need guidance during return to running activity as restrictions are lifted due to less knowledge on safe training increases [21]. Clinicians may use patients' data to objectively inform remote interventions to promote health and decrease injury risk. Overall, the survey outcomes support telemedicine initiatives in the upcoming months to years to conduct remote monitoring and interventional plans for runners in training and recovery contexts.

## Limitations

This was a cross-sectional survey of running behaviors, and therefore there is a potential for recall bias in responses pertaining to previous training habits and injury history. We attempted to account for this issue by asking respondents about the year prior to the pandemic, and not a longer timeframe. Given that social isolation measures occurred at different timepoints globally, we asked participants to respond in the timeframe that they experienced these protective prevention measures. For this reason, we could not calculate true injury risk due to differences in assessment timeframes. However, we decided to perform an injury risk adjustment given the social isolation procedures were in effect for the majority of respondents in the United States for three months (March-May). We believe the equivocal time period comparison most appropriately modeled the observed effects. Future work exploring injury risk during the return to routine functioning on a similar timeframe to the year prior to social isolation to get a true estimate of RRI following the pandemic is warranted. We were unable to assess re-injuries or chronic injuries explicitly in this study which may have influenced the RRI findings, however we accounted for previous injury as a covariate in regression analyses. Running goals,

for example, recreational versus elite participation, is likely to influence motivation. While we did assess runners' motives for running, we did not obtain runners' primary running level and could not perform sub-group analyses to this point. Finally, the survey was only distributed in English, and the majority of respondents were from the United States, and primarily on the east coast. The results may have been influenced more heavily by pandemic response measures in these regions, and therefore the results should be interpreted as descriptive outcomes for the included sample and may not reflect the global running community.

## Conclusions

The COVID-19 pandemic influenced runners' behaviors and resulted in increased training volume with decreased training intensity. Runners' motivations for running overall declined, and shifted from competition and socialization towards fitness, stress relief, and occupying time. Running-related injury risk was overall higher during the pandemic for lower extremity overuse injuries compared to the year prior. These findings highlight changes in running training patterns, motivations, and injury risk in adult distance runners and should be considered by coaches and sports medicine clinicians as social isolation measures are relaxed.

## Supporting information

**S1 Fig. (A) Percentage of respondents by state within the United States, (B) percentage of respondents by country, (C) responses by state: change in number of sustained runs, (D) responses by country: change in number of sustained runs, (E) responses by state: change in number of running workouts, (F) responses by country: change in number of running workouts.** Abbreviations: Avg, Average; Diff, Difference; N, number.
(TIF)

**S1 File. Running behaviors before and during the COVID-19 pandemic survey.** List of questions and response options included in the survey. Question logic was used to skip irrelevant questions and displayed within the document.
(DOCX)

**S1 Table. Open coding short response items.** The words or phrases that most frequently occurred in short responses were grouped into like categories and displayed in the table, along with number of instances noted throughout the short responses.
(DOCX)

## Acknowledgments

We would like to thank members of the Exercise and Sport Injury Laboratory for their feedback on this survey, and Luzita Vela for her guidance in qualitative analyses.

## Author Contributions

**Conceptualization:** Alexandra F. DeJong, Jay Hertel.

**Data curation:** Alexandra F. DeJong, Pamela N. Fish.

**Formal analysis:** Alexandra F. DeJong, Pamela N. Fish, Jay Hertel.

**Investigation:** Alexandra F. DeJong.

**Project administration:** Jay Hertel.

**Supervision:** Jay Hertel.

**Visualization:** Alexandra F. DeJong, Pamela N. Fish.

**Writing – original draft:** Alexandra F. DeJong.

**Writing – review & editing:** Pamela N. Fish, Jay Hertel.

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
