## [Decision Letter · Decision Letter 0]

15 Jul 2020

PONE-D-20-18783

Running behaviors, motivations, and injury risk during the COVID-19 pandemic: A survey of 1147 international runners

PLOS ONE

Dear Dr. DeJong,

Thank you for submitting your manuscript to PLOS ONE. After careful consideration, we feel that it has merit but does not fully meet PLOS ONE’s publication criteria as it currently stands. Therefore, we invite you to submit a revised version of the manuscript that addresses the points raised during the review process.

Two experts in the field reviewd your manuscript and, although they found merit, they have several concerns. First of all on the definition of injury and running related injury, and secondly, on the sampled population and the duration of lockdown in each country (i-.e complete lockdown? partial lockdown, for how long did each country remain in quarantine?)

We look forward to receiving your revised manuscript.

Kind regards,

Maria Francesca Piacentini

Academic Editor

PLOS ONE

Journal Requirements:

2.In your Data Availability statement, you have not specified where the minimal data set underlying the results described in your manuscript can be found. PLOS defines a study's minimal data set as the underlying data used to reach the conclusions drawn in the manuscript and any additional data required to replicate the reported study findings in their entirety. All PLOS journals require that the minimal data set be made fully available. For more information about our data policy, please see http://journals.plos.org/plosone/s/data-availability.

3. Please ensure that you include a title page within your main document. You should list all authors and all affiliations as per our author instructions and clearly indicate the corresponding author.

4.We note that [Figure(s) 1, and S1(A,B,C,D,E,F)] in your submission contain [map/satellite] images which may be copyrighted. All PLOS content is published under the Creative Commons Attribution License (CC BY 4.0), which means that the manuscript, images, and Supporting Information files will be freely available online, and any third party is permitted to access, download, copy, distribute, and use these materials in any way, even commercially, with proper attribution. For these reasons, we cannot publish previously copyrighted maps or satellite images created using proprietary data, such as Google software (Google Maps, Street View, and Earth). For more information, see our copyright guidelines: http://journals.plos.org/plosone/s/licenses-and-copyright.

1.    You may seek permission from the original copyright holder of Figure(s) [1, and S1(A,B,C,D,E,F)] to publish the content specifically under the CC BY 4.0 license. 

5. Please include your tables as part of your main manuscript and remove the individual files. Please note that supplementary tables (should remain/ be uploaded) as separate "supporting information" files.

6. Please upload a copy of Supplementary Table which you refer to in your text on page 14.

7. Please include captions for your Supporting Information files at the end of your manuscript, and update any in-text citations to match accordingly. Please see our Supporting Information guidelines for more information: http://journals.plos.org/plosone/s/supporting-information

Reviewers' comments:

Reviewer's Responses to Questions

**Comments to the Author**

1. Is the manuscript technically sound, and do the data support the conclusions?

Reviewer #1: Partly

Reviewer #2: Partly

2. Has the statistical analysis been performed appropriately and rigorously? 

Reviewer #1: Yes

Reviewer #2: No

3. Have the authors made all data underlying the findings in their manuscript fully available?

Reviewer #1: No

Reviewer #2: Yes

4. Is the manuscript presented in an intelligible fashion and written in standard English?

Reviewer #1: Yes

Reviewer #2: Yes

5. Review Comments to the Author

Reviewer #1: Thank you for your interesting and timely paper. You have obviously put a great deal of hard work, over a short period of time, into this study, for which I congratulate you. I should like to make a few suggestions, however, as to how (I feel) you might be able to improve the usefulness to others, as well as the and citability, of your work:-

1. Did you define injury in the survey? If so, please include the definition in the paper.

2. Did you also differentiate between overuse and traumatic injury in the survey (or only between different injury types as per Table 3)? If you did not your report of the strength of the links between the injury data and the training modification related data could be misleading. Please clarify what you did.

3. Was the survey in English only (seeing as this potentially impact on the validity of the answers to it from some countries) ? I presume so – if this was the case did you in fact assess the extent to which your respondents were native English speakers? I supposed from Fig 1A that they were but then Fig 1ab and line 209-213 seem to show that they may not all have been so. It would have helped me assess your paper if I could have seen the survey- it sounds from what you wrote in Line 94 that you intended to include it in an appendix but there was no appendix in what I received

4. Did you assess how long the subjects has been in lock down and the extent to that lockdown in the survey? You seem to assume that it had been for 3 months but you collected the data over a period of 1 month so the lockdown could have been over up to 4 in some cases and this would affect your calculations

5. Why did you ask for training data for the previous year rather than for say the three months prior to lockdown (for which recall would have been better) or regarding “what you would normally do at this period of your competitive/training year”? You report training data to 2 decimal places but presumably (and from my experience and data) the number and volume of sessions changes from one type of macrocycle to the next (e.g. between pre-competition and competition or endurance base and pre-competition periods). The extent to which it does so may well be more than the pre-during pandemic changes that you report, and render your method of assessing pandemic induced change (ie subtract one from the other) less meaningful than it might have been.

6. I realise that you have loads of data but the way you present the data in the figures makes it difficult to see exactly what they are. Fig 3AB for example is hard to understand- perhaps you might consider whether there is a better way of showing these data?

7. Your supplemental data seem to show that you used varying numbers of respondents for the statistical analysis- as the number of responses to each question differed at different points of the survey? Was this the case? In line 135 you state that only complete data were used

8. Some of the legends are incomplete e.g. for Table 2 you mention EC vs WC but I could not see the explanation of the abbreviation, in Table 1 you do not specific units for mileage

9. From your Table 3 and using the reasoning that you proposed/used elsewhere (divide the values for a year by 4, if I understood that correctly) the injury n was higher for both sprains and strains before the pandemic than it was during (30 vs. 27 and 81.75 vs. 80), apart from in the case of fractures (14.5 vs 8). This isn’t discussed. It contradicts your conclusion in the abstract. Unless you did not in fact analyse complete data sets as noted in the paper? (or I have misunderstood in which please accept my apologies for this point).

10. Some of your discussion goes over stuff that you have not included in the results. I would revisit both sections accordingly.

Thank you in advance for the clarifications.

Best wishes

Reviewer #2: This manuscript investigates the effect of COVID-19 on running behaviors and motivation, using an international survey.

I found the paper to be generally well written. The manuscript addresses a very relevant and important topic. Overal, the methods section is detailed, although some important remarks need to be considered to strengthen the paper. The results are clear and comprehensive. However, some concerns relating to the statistical analysis and interpretations of the results are present, which should be revised before considering publication.

2.1. General comments:

1. The main concern relates to the representativeness of the survey. A convenience sample was used in the survey, without prior calculation of the number of participants needed for the survey to be representative for the targeted population (i.e. international runners). The manuscript should be revised while integrating this reflection, both in the methods, results and discussion. This prevents the interested reader from misinterpreting the results and providing a sound answer to the question: "Are these results able to represent the behavior of the complete population (if yes, include total error of the survey), or are these results to be interpreted as descriptive for the included sample."

2. Data-analysis: the authors select the paired t-test, without description whether or not the required conditions were fulfilled. Also, please provide more detail in the survey analysis (i.e. depending on data level, processing of open answers)

3. Data-analysis: the authors select to adjust injury data at group level and not at individual level. Did the survey not allow to calculate or estimate individual exposure and injury risk? Please provide additional rationale for this extrapolation at group level, because this influences greatly the results.

Additional comments:

- Injury definition of running related injuries: please elaborate on how recall bias influences injury data throughout the survey. Overuse injuries (running related injuries) are susceptible for this kind of bias and are dependent of the selected injury definition.

- The survey aims to include international runners, but the COVID-quarantine differs in each country / region. This conflicts with earlier comment on representativeness. Consider sub-group analysis for those regions that allow for a more representative analysis?

- The survey included all levels of runners, but the level (competition or recreational) can be of great influence to motivation. Group analysis might conceal sub-group differences? Did the authors consider sub-group analysis for motivation?

- How reliable are the exposure, injury and training data? Can the authors provide some additional insights in this matter?

- Results: what about the re-injuries or chronic injuries? How were these processed during data-analysis?

- Discussion: The decreased intensity seems to contradict the increased injury risk? This is to be interpreted in the broader context of training volume; including both training intensity and frequency, so please discus this interaction more extensively.

- Discussion: see above, please reflect on the representativeness and/or margin of error of the survey, relating to either the entire population or the included sample.

6. PLOS authors have the option to publish the peer review history of their article (what does this mean?). If published, this will include your full peer review and any attached files.

Reviewer #1: No

Reviewer #2: No

---

## [Author Response · Author response to Decision Letter 0]

4 Aug 2020

Responses to Editor

Response: We have made the appropriate stylistic adjustments, and highlighted the changes throughout the revised manuscript.

2.In your Data Availability statement, you have not specified where the minimal data set underlying the results described in your manuscript can be found. PLOS defines a study's minimal data set as the underlying data used to reach the conclusions drawn in the manuscript and any additional data required to replicate the reported study findings in their entirety. All PLOS journals require that the minimal data set be made fully available. For more information about our data policy, please see http://journals.plos.org/plosone/s/data-availability.

Response: There is an ethical restriction to sharing the data publicly, as the IRB-approved study consent form had a required section that indicated to participants that their data will be protected, will not be shared, and will be destroyed upon study completion. We do not have contact information for all participants since this was an anonymous online survey and cannot follow-up asking if they would be willing to share their data. We have been in contact with the IRB, and they informed us that subsequently, the data cannot be shared publicly. Instead, we have attached the output from the statistical analyses for increased data transparency.

3. Please ensure that you include a title page within your main document. You should list all authors and all affiliations as per our author instructions and clearly indicate the corresponding author.

Response: We have included a title page now within the main document.

4.We note that [Figure(s) 1, and S1(A,B,C,D,E,F)] in your submission contain [map/satellite] images which may be copyrighted. All PLOS content is published under the Creative Commons Attribution License (CC BY 4.0), which means that the manuscript, images, and Supporting Information files will be freely available online, and any third party is permitted to access, download, copy, distribute, and use these materials in any way, even commercially, with proper attribution. For these reasons, we cannot publish previously copyrighted maps or satellite images created using proprietary data, such as Google software (Google Maps, Street View, and Earth). For more information, see our copyright guidelines: http://journals.plos.org/plosone/s/licenses-and-copyright.

1. You may seek permission from the original copyright holder of Figure(s) [1, and S1(A,B,C,D,E,F)] to publish the content specifically under the CC BY 4.0 license. 

Response: We contacted Mapbox, and received the following information:

Thanks for writing in. While we do not sign paperwork for special agreements, our Terms of Service serves as the contract for use of your Mapbox account. As long as your maps follow our attribution and watermark requirements, you can use them in your publication.

While we could not obtain the signed copyright form, we adjusted the images according to their requirements and should be suitable for publication.

5. Please include your tables as part of your main manuscript and remove the individual files. Please note that supplementary tables (should remain/ be uploaded) as separate "supporting information" files.

Response: We have included the tables now in the main manuscript and removed the individual files.

6. Please upload a copy of Supplementary Table which you refer to in your text on page 14.

Response: We apologize the table did not successfully upload in the original submission, we have re-formatted the document and included this now in the revision.

7. Please include captions for your Supporting Information files at the end of your manuscript, and update any in-text citations to match accordingly. Please see our Supporting Information guidelines for more information: http://journals.plos.org/plosone/s/supporting-information

Response: This information is now included at the end of the manuscript.

Responses to Reviewer 1

Reviewer #1: Thank you for your interesting and timely paper. You have obviously put a great deal of hard work, over a short period of time, into this study, for which I congratulate you. I should like to make a few suggestions, however, as to how (I feel) you might be able to improve the usefulness to others, as well as the and citability, of your work:-

Response: Thank you for taking the time to review our manuscript, we appreciate the insights intended to strengthen the interpretation of the survey findings. We have addressed the specific comments and questions below to the best of our ability. 

1. Did you define injury in the survey? If so, please include the definition in the paper.

Response: We kept the injury inclusion for this study broad, and aimed to capture more specific injury information about the injury types by body part in the table included in the survey. We apologize, the PDF of the survey did not successfully upload in the initial submission. We re-formatted this supplemental material and uploaded the details in this revision. We have address this in the text on lines 116-117. 

2. Did you also differentiate between overuse and traumatic injury in the survey (or only between different injury types as per Table 3)? If you did not your report of the strength of the links between the injury data and the training modification related data could be misleading. Please clarify what you did.

Response: We did not specifically differentiate between overuse and traumatic injuries, however we did collection information on the duration that the injury influenced their running training, as well as time off from injury. Since we also collected information on injury types, we believe that we still have collected relevant data that provides insight into overuse versus acute injuries. We included this point on lines 369-375, and 460-462.

3. Was the survey in English only (seeing as this potentially impact on the validity of the answers to it from some countries) ? I presume so – if this was the case did you in fact assess the extent to which your respondents were native English speakers? I supposed from Fig 1A that they were but then Fig 1ab and line 209-213 seem to show that they may not all have been so. It would have helped me assess your paper if I could have seen the survey- it sounds from what you wrote in Line 94 that you intended to include it in an appendix but there was no appendix in what I received

Response: The survey was only provided in English, we have included a statement about this now in the methods on line 84, and acknowledged this as a limitation on lines 465-466. We apologize that the survey did not successfully upload in the initial submission, this is available now as a supplemental document.

4. Did you assess how long the subjects has been in lock down and the extent to that lockdown in the survey? You seem to assume that it had been for 3 months but you collected the data over a period of 1 month so the lockdown could have been over up to 4 in some cases and this would affect your calculations

Response: We did not explicitly assess the length/extent of lock-down in this survey. Unfortunately, this information is extremely specific to individuals and to the region in which they live. We decided to utilize the average amount of time the quarantine was in effect. We included this information on lines 452-458 as a limitation.

5. Why did you ask for training data for the previous year rather than for say the three months prior to lockdown (for which recall would have been better) or regarding “what you would normally do at this period of your competitive/training year”? You report training data to 2 decimal places but presumably (and from my experience and data) the number and volume of sessions changes from one type of macrocycle to the next (e.g. between pre-competition and competition or endurance base and pre-competition periods). The extent to which it does so may well be more than the pre-during pandemic changes that you report, and render your method of assessing pandemic induced change (ie subtract one from the other) less meaningful than it might have been.

Response: We decided to use the year prior to the pandemic to get a representative sample of running over time as opposed to the influence of fluctuations in training. We added this information on lines 95-96.

Another reason why we opted to use the year time-frame is because we will be sending out a follow-up survey in a years’ time to assess running outcomes and injury data. In this way, we will now have equivocal data collection time frames to assess change in injury risk following return to activity from the pandemic. 

6. I realise that you have loads of data but the way you present the data in the figures makes it difficult to see exactly what they are. Fig 3AB for example is hard to understand- perhaps you might consider whether there is a better way of showing these data?

Response: We apologize that this image was not clear. We changed the image to a histogram of the responses to better represent the data.

7. Your supplemental data seem to show that you used varying numbers of respondents for the statistical analysis- as the number of responses to each question differed at different points of the survey? Was this the case? In line 135 you state that only complete data were used

Response: Only complete data was used; we had an optional response section for the short-responses, so if this section was not completed, we still included that participant as a complete response. For injury data, we assessed proportions of injuries based on the injured participant pool which may reflect a smaller sample. For the supplemental figures, these were used to display the average differences in running behaviors by region in which the entire sample was assessed and displayed. We did not include international respondents in the regression analysis since there was not a sufficient number of respondents in each geographic group and therefore the interpretations would not be appropriate.

8. Some of the legends are incomplete e.g. for Table 2 you mention EC vs WC but I could not see the explanation of the abbreviation, in Table 1 you do not specific units for mileage

Response: We apologize for this omission; we have fixed the legends for the tables accordingly in Table 1, and on line 271 for the caption for Table 2.

9. From your Table 3 and using the reasoning that you proposed/used elsewhere (divide the values for a year by 4, if I understood that correctly) the injury n was higher for both sprains and strains before the pandemic than it was during (30 vs. 27 and 81.75 vs. 80), apart from in the case of fractures (14.5 vs 8). This isn’t discussed. It contradicts your conclusion in the abstract. Unless you did not in fact analyse complete data sets as noted in the paper? (or I have misunderstood in which please accept my apologies for this point).

Response: We apologize, the entire sample was used, there was a typo when entering in the reported percentage of reported injuries in context to the entire sample in the text, the percentage is now fixed on line 282. For additional clarity in the subsequent percentage break-downs in Table 3, we looked at the proportion of injured respondents and assessed injury type breakdown within the injured pool. The overall injury percentage was taken as a part of the entire sample; however, the injury type/location proportion was based on the injury sample. We have now explicitly explained this in the text on lines 173-174. Although the number of injuries was higher, the proportion of these injuries to adjust for time was higher during the pandemic. We believe that the differences in the raw numbers are attributed to the time frame of exposure, which is a delimitation to this study and why we only discussed proportions in the manuscript. We have now explained this point on lines 287-288.

10. Some of your discussion goes over stuff that you have not included in the results. I would revisit both sections accordingly.

Response: Thank you for bringing this to our attention, we have made appropriate changes on lines 149, and 204.

Thank you in advance for the clarifications.

Best wishes

Thank you very much again for taking the time to review this manuscript.

Responses to Reviewer 2

Reviewer #2: This manuscript investigates the effect of COVID-19 on running behaviors and motivation, using an international survey.

I found the paper to be generally well written. The manuscript addresses a very relevant and important topic. Overall, the methods section is detailed, although some important remarks need to be considered to strengthen the paper. The results are clear and comprehensive. However, some concerns relating to the statistical analysis and interpretations of the results are present, which should be revised before considering publication.

Response: We would like to thank you for taking the time to review our manuscript, and we appreciate the comments and suggestions to improve the paper. We have addressed the specific concerns below to the best of our ability.

2.1. General comments:

1. The main concern relates to the representativeness of the survey. A convenience sample was used in the survey, without prior calculation of the number of participants needed for the survey to be representative for the targeted population (i.e. international runners). The manuscript should be revised while integrating this reflection, both in the methods, results and discussion. This prevents the interested reader from misinterpreting the results and providing a sound answer to the question: "Are these results able to represent the behavior of the complete population (if yes, include total error of the survey), or are these results to be interpreted as descriptive for the included sample."

Response: We agree that based on the sample of convenience that we cannot extrapolate these outcomes to the running community at large. We included this notion in the methods on line 123, results on line 200, and discussion on lines 324-325, 398-399, and 466-469. We also adjusted our title so as not to mislead the audience given that the majority of responses were from the US.

2. Data-analysis: the authors select the paired t-test, without description whether or not the required conditions were fulfilled. Also, please provide more detail in the survey analysis (i.e. depending on data level, processing of open answers)

Response: Thank you for raising this concern, we assessed the data for normality, the sample data were numeric and continuous, and the t-test was most appropriate since the outcomes were collected for the same participants just assessing at different timepoints (repeated measures). We included a statement about this in the statistical analysis section now on lines 149-151, and 153-154. We also provided additional details about the open coding process on lines 187-189.

3. Data-analysis: the authors select to adjust injury data at group level and not at individual level. Did the survey not allow to calculate or estimate individual exposure and injury risk? Please provide additional rationale for this extrapolation at group level, because this influences greatly the results.

Response: Unfortunately we were unable to adjust the data at the individual level since we did not take the length of time that the participants were in quarantine in their region in the survey, and since social isolation measures were extremely specific to exact geographical location (even more than, for example, the state level in the US). Therefore, we decided to adjust the injury data at the group level. However, we will be conducting a follow-up study one year after the original survey was sent out to assess injury data to get a more sensitive estimate of injury data following the pandemic as the time frames will be the same as the year before the pandemic to the year following. We included this limitation on lines 452-458.

Additional comments:

- Injury definition of running related injuries: please elaborate on how recall bias influences injury data throughout the survey. Overuse injuries (running related injuries) are susceptible for this kind of bias and are dependent of the selected injury definition.

Response: We agree this is an important area to highlight in the manuscript. We kept the injury inclusion for this study broad, and aimed to capture more specific injury information about the injury types by body part in the table included in the survey. We did not specifically differentiate between overuse and traumatic injuries, however we did collection information on the duration that the injury influenced their running training, as well as time off from injury. We included this information in the methods (lines 116-117), discussion (lines 369-375), and acknowledged in the limitations (lines 460-462).

- The survey aims to include international runners, but the COVID-quarantine differs in each country / region. This conflicts with earlier comment on representativeness. Consider sub-group analysis for those regions that allow for a more representative analysis?

Response: While we did have international respondents for this study, we were not able to include international respondents as a sub-group in the regression analysis since there was not a sufficient number of respondents in each geographic group and therefore the interpretations would not be appropriate (lines 398-399). Instead, we included the breakdown of responses by geographical location, and changes in running outcomes by geographical location in the supplemental material. We also adjusted the title as noted in a previous response to address this matter.

- The survey included all levels of runners, but the level (competition or recreational) can be of great influence to motivation. Group analysis might conceal sub-group differences? Did the authors consider sub-group analysis for motivation?

Response: We certainly agree that running level can influence motivations. Unfortunately, there was not an explicit question asking what the runners’ primary reason for participation was, instead we asked that runners selected all reasons that applied for their motives for running training (please see supplemental survey included in this revision). We did determine from this questions that 200 respondents (17%) indicated they did compete/race, however only 23 solely responded that their motives for running were competition alone without recreational motives also selected. Therefore, there was not a sufficient sample to assess this influence directly. Instead, we looked at the influence of experience in the regression analysis. We noted this notion as a limitation on lines 462-465. 

- How reliable are the exposure, injury and training data? Can the authors provide some additional insights in this matter?

Response: We acknowledge that these data are influenced by recall bias given the nature of the study. We believe that the exposure/training data are reliable given that 90% of our sample used technology to track their runs, and therefore had concrete data to support their mileage and training behaviors. For the injury data, we relied on self-report from participants. Since we included the year prior to the study and only injuries that were resultant from running training, we attempted to minimize recall bias and capture relevant injury data. We did acknowledge these limitations in the manuscript on lines 449-450. 

- Results: what about the re-injuries or chronic injuries? How were these processed during data-analysis?

Response: We did not explicitly ask if injuries were re-injuries or chronic injuries in the survey, however we did covary for previous injury in the regression analysis to determine the influence of previous/recurrent injury on predicting injury during the pandemic, however the regression model was not significant (lines 290-291). We did obtain the length of time taken off from injury, which was listed on lines 283-285. This may provide some additional injury insights. We also did assess for injury types, and typically the bony injuries and injuries in the “other” categories were overuse injury types (i.e. table 3, lower limb bony injuries were stress fractures, and the other categories for knee and lower leg were primarily patellofemoral pain and medial tibial stress syndrome respectively). We included this information in the manuscript on lines 369-375. We acknowledged re-injury and injury chronicity as a limitation on lines 460-462.

- Discussion: The decreased intensity seems to contradict the increased injury risk? This is to be interpreted in the broader context of training volume; including both training intensity and frequency, so please discuss this interaction more extensively.

Response: Thank you for raising this point; our theory is that runners opted to increase total running distance and volume, thereby increasing their exposure to loading and potentially increasing the risk of bony/overuse injuries. While increasing intensity has been related to increased acute injuries, we believed that decreasing typical running intensity would actually adversely affect chronic training adaptations, specifically with muscle tone and protection against lower extremity stress-related injuries. Therefore, if runners were transitioning from typical interval training to adding more long-duration higher volume runs, this would be a contributing factor to higher injury rates. We explained this further in the manuscript discussion on lines 345-348.

- Discussion: see above, please reflect on the representativeness and/or margin of error of the survey, relating to either the entire population or the included sample.

Response: Based on previous comments, we have adjusted the discussion accordingly on lines 324-325, 398-399, and in the limitations on lines 468-469.

Thank you again for taking the time to review this manuscript.

---

## [Decision Letter · Decision Letter 1]

30 Sep 2020

PONE-D-20-18783R1

Running behaviors, motivations, and injury risk during the COVID-19 pandemic: A survey of 1147 runners

PLOS ONE

Dear Dr. DeJong,

Thank you for submitting your manuscript to PLOS ONE. After careful consideration, we feel that it has merit but does not fully meet PLOS ONE’s publication criteria as it currently stands. Therefore, we invite you to submit a revised version of the manuscript that addresses the points raised during the review process.

There are still minor issues that the reviewers would like to see addressed

We look forward to receiving your revised manuscript.

Kind regards,

Maria Francesca Piacentini

Academic Editor

PLOS ONE

Reviewers' comments:

Reviewer's Responses to Questions

**Comments to the Author**

1. If the authors have adequately addressed your comments raised in a previous round of review and you feel that this manuscript is now acceptable for publication, you may indicate that here to bypass the “Comments to the Author” section, enter your conflict of interest statement in the “Confidential to Editor” section, and submit your "Accept" recommendation.

Reviewer #1: (No Response)

Reviewer #2: All comments have been addressed

2. Is the manuscript technically sound, and do the data support the conclusions?

Reviewer #1: Partly

Reviewer #2: Partly

3. Has the statistical analysis been performed appropriately and rigorously? 

Reviewer #1: Yes

Reviewer #2: No

4. Have the authors made all data underlying the findings in their manuscript fully available?

Reviewer #1: Yes

Reviewer #2: Yes

5. Is the manuscript presented in an intelligible fashion and written in standard English?

Reviewer #1: Yes

Reviewer #2: Yes

6. Review Comments to the Author

Reviewer #1: Thank you for your revised paper and replies to my comments.

A few points:

Line 10, 152, 205. Student´s t test

L13, 284. Sentences should never start with a number in numerical format.

L52, 165, 191, 206, 377 Something wrong with the text- part of it seems to be cut off

L56 moulding?

L73, 88, 89, 116 etc RRI´s (you left out the apostrophe)

L130-131, 132, 196-198, 301 “participants residing in all countries”, “through June”, “about participants’ perceived impact the pandemic had on”, “participants that” are ungrammatical

L155 as compared to

L178 “put into in the model” needs to be corrected

Table 1 units incomplete (e.g. non for pace)

I think you should explicitly state that injury was not defined in the survey

Thank you.

Reviewer #2: Thank you for your thorough revision of the manuscript and your honest and clear responses. Although the study protocol and collected data did not allow all concerns to be fully remediated, the authors have listed these as a study limitation and revised the manuscript and conclusions accordingly whenever possible. I think the editor is now best suited to evaluate the manuscript. If the manuscript still fits the scope and policy of Plos One, I could consent with accepting this manuscript for publication.

7. PLOS authors have the option to publish the peer review history of their article (what does this mean?). If published, this will include your full peer review and any attached files.

Reviewer #1: No

Reviewer #2: No

---

## [Author Response · Author response to Decision Letter 1]

30 Sep 2020

Reviewer #1: Thank you for your revised paper and replies to my comments. 

Response: We would like to thank Reviewer 1 for taking the time to review the revised manuscript. We addressed all specific comments below.

A few points:

Line 10, 152, 205. Student´s t test 

Response: Paired t-tests were changed to Student’s t-tests on lines 10, 153, 166, 206, 213.

L13, 284. Sentences should never start with a number in numerical format. 

Response: The sentence structure was adjusted on lines 13 and 284.

L52, 165, 191, 206, 377 Something wrong with the text- part of it seems to be cut off Response: We apologize that the text was cut off for the uploaded version, we are unsure what happened as we checked to make sure everything looked in order on the submission. There does not appear to be any truncated text on this modification, we apologize again for the error.

L56 moulding? 

Response: We revised “molding” to “influencing” to enhance clarity.

L73, 88, 89, 116 etc RRI´s (you left out the apostrophe) Response: All instances of “RRIs” have been fixed to “RRI’s”

L130-131, 132, 196-198, 301 “participants residing in all countries”, “through June”, “about participants’ perceived impact the pandemic had on”, “participants that” are ungrammatical Response: We edited each sentence to be grammatically correct.

L155 as compared to 

Response: We added “as” to the sentence on line 155.

L178 “put into in the model” needs to be corrected Response: We adjusted this to now read “and put into the model” on line 179.

Table 1 units incomplete (e.g. non for pace) 

Response: We added in all units now in Table 1.

I think you should explicitly state that injury was not defined in the survey Response: We added this information on line 117.

Thank you.

Reviewer #2: Thank you for your thorough revision of the manuscript and your honest and clear responses. Although the study protocol and collected data did not allow all concerns to be fully remediated, the authors have listed these as a study limitation and revised the manuscript and conclusions accordingly whenever possible. I think the editor is now best suited to evaluate the manuscript. If the manuscript still fits the scope and policy of Plos One, I could consent with accepting this manuscript for publication. 

Response: We would like to thank Reviewer 2 for taking the time to review the revised manuscript, and for your feedback and comments throughout this process.

---

## [Decision Letter · Decision Letter 2]

18 Jan 2021

Running behaviors, motivations, and injury risk during the COVID-19 pandemic: A survey of 1147 runners

PONE-D-20-18783R2

Dear Dr. DeJong,

We’re pleased to inform you that your manuscript has been judged scientifically suitable for publication and will be formally accepted for publication once it meets all outstanding technical requirements.

Kind regards,

Daniel Boullosa

Academic Editor

PLOS ONE

Additional Editor Comments (optional):

Reviewers' comments:

Reviewer's Responses to Questions

**Comments to the Author**

1. If the authors have adequately addressed your comments raised in a previous round of review and you feel that this manuscript is now acceptable for publication, you may indicate that here to bypass the “Comments to the Author” section, enter your conflict of interest statement in the “Confidential to Editor” section, and submit your "Accept" recommendation.

Reviewer #1: All comments have been addressed

Reviewer #2: All comments have been addressed

2. Is the manuscript technically sound, and do the data support the conclusions?

Reviewer #1: Partly

Reviewer #2: Partly

3. Has the statistical analysis been performed appropriately and rigorously? 

Reviewer #1: Yes

Reviewer #2: Yes

4. Have the authors made all data underlying the findings in their manuscript fully available?

Reviewer #1: Yes

Reviewer #2: Yes

5. Is the manuscript presented in an intelligible fashion and written in standard English?

Reviewer #1: Yes

Reviewer #2: Yes

6. Review Comments to the Author

Reviewer #1: (No Response)

Reviewer #2: (No Response)

7. PLOS authors have the option to publish the peer review history of their article (what does this mean?). If published, this will include your full peer review and any attached files.

Reviewer #1: No

Reviewer #2: No

---

## [Editor Report · Acceptance letter]

25 Jan 2021

PONE-D-20-18783R2 

Running behaviors, motivations, and injury risk during the COVID-19 pandemic: A survey of 1147 runners 

Dear Dr. DeJong:

I'm pleased to inform you that your manuscript has been deemed suitable for publication in PLOS ONE. Congratulations! Your manuscript is now with our production department. 

Kind regards, 

on behalf of

Dr. Daniel Boullosa 

Academic Editor

PLOS ONE